COMMUNICATIONS

# Magnetic field controlled charge density wave coupling in underdoped $YBa_2Cu_3O_{6+x}$

J. Chang[1], E. Blackburn[2], O. Ivashko[1], A.T. Holmes[3], N.B. Christensen[4], M. Hücker[5], Ruixing Liang[6,7], D.A. Bonn[6,7], W.N. Hardy[6,7], U. Rütt[8], M.v. Zimmermann[8], E.M. Forgan[2] & S.M. Hayden[9]

The application of magnetic fields to layered cuprates suppresses their high-temperature superconducting behaviour and reveals competing ground states. In widely studied underdoped $YBa_2Cu_3O_{6+x}$ (YBCO), the microscopic nature of field-induced electronic and structural changes at low temperatures remains unclear. Here we report an X-ray study of the high-field charge density wave (CDW) in YBCO. For hole dopings $\sim 0.123$, we find that a field ($B \sim 10\,T$) induces additional CDW correlations along the CuO chain ($b$-direction) only, leading to a three-dimensional (3D) ordered state along this direction at $B \sim 15\,T$. The CDW signal along the $a$-direction is also enhanced by field, but does not develop an additional pattern of correlations. Magnetic field modifies the coupling between the $CuO_2$ bilayers in the YBCO structure, and causes the sudden appearance of the 3D CDW order. The mirror symmetry of individual bilayers is broken by the CDW at low and high fields, allowing Fermi surface reconstruction, as recently suggested.

[1] Physik-Institut, Universität Zürich, Winterthurerstrasse 190, Zürich CH-8057, Switzerland. [2] School of Physics and Astronomy, University of Birmingham, Birmingham B15 2TT, UK. [3] European Spallation Source ERIC, Box 176, Lund SE-221 00, Sweden. [4] Department of Physics, Technical University of Denmark, Kongens Lyngby DK-2800, Denmark. [5] Condensed Matter Physics & Materials Science Department, Brookhaven National Lab, Upton, New York 11973, USA. [6] Department of Physics & Astronomy, University of British Columbia, Vancouver V6T-1Z1, Canada. [7] Canadian Institute for Advanced Research, Toronto M5G-1Z8, Canada. [8] Deutsches Elektronen-Synchrotron DESY, 22603 Hamburg, Germany. [9] H.H. Wills Physics Laboratory, University of Bristol, Bristol BS8 1TL, UK. Correspondence and requests for materials should be addressed to J.C. (email: johan.chang@physik.uzh.ch) or to S.M.H. (email: s.hayden@bristol.ac.uk).

Charge density wave (CDW) correlations[1], that is, periodic modulations of the electronic charge density accompanied by a periodic distortion of the atomic lattice, have long been known to exist in underdoped La-based cuprate high-temperature superconductors[2,3]. More recently, it has been found that charge order is a universal property of underdoped high-temperature cuprate superconductors[4–11]. CDW correlations appear typically at temperatures well above the superconducting transition temperature $T_c$. Cooling through $T_c$ suppresses the CDW and leads to a state, in which the superconducting and CDW order parameters are intertwined and competing[12–14].

The application of magnetic fields suppresses super-conductivity. In the case of underdoped $YBa_2Cu_3O_{6+x}$ (YBCO), a number of changes in electronic properties have been reported in the field range $B \approx 10$–20 T. For example, new splittings occur in NMR spectra[11,15], ultrasound shows anomalies in the elastic constants[16] and the thermal Hall effect suggests that there is an electronic reconstruction[17]. At larger fields, $B \gtrsim 25$ T a normal state with quantum oscillations (QO)[18] and coherent transport along the $c$ axis[19] is observed. The existence of QO, combined with a high-field negative Hall and Seebeck effect, is most easily understood in terms of electron pockets[9,20–23].

Fields $B \approx 10$–20 T also cause changes in the CDW order that can be seen by X-ray measurements. Initial experiments[5] showed that a magnetic field causes an enhancement of the diffuse CDW scattering[5,8]. A recent X-ray free-electron laser experiment[24] has shown that a magnetic field of $B \gtrsim 15$ T induces a new CDW Bragg peak, with a propagation vector along the $b$ axis, corresponding to an extended range of ordering along the $c$ axis and an in-phase correlation of the CDW modulation between the neighbouring bilayers.

It is important to determine the nature of the CDW correlations induced by the magnetic field in YBCO and their relationship to the electronic properties. Of particular interest are the high-field CDW phase diagram and whether a field also induces new CDW order propagating along the $a$ axis. We have therefore used hard X-ray scattering measurements to determine the evolution of the CDW correlations, with magnetic fields up to 16.9 T for several doping levels. Here we investigate the CDW for propagation vectors along the crystallographic $a$- and $b$-directions, allowing us to extend the pulsed-field measurements[24] and identify new field-induced anisotropies in the CDW. By measuring the profile of the diffuse CDW scattering as a function of field, we show that the CDW inter-bilayer coupling along the $c$ axis is strongly field dependent. We also show that field-induced changes in the CDW can be associated with many of the anomalies[11,15–17,25] observed in electronic properties. In particular, the $B - T$ phase diagram has two boundary lines associated with the formation of high-field CDW order. Our data also provides insight into the likely high-field structure of the CDW (in the normal state) that is relevant to describe the Fermi surface reconstruction leading to QO.

## Results

### Charge density wave order in YBCO.
The CDW correlations in the cuprates have propagation vectors with the in-plane components parallel to the Cu–O bonds and periodicities of $3 - 4a$ depending on the system[2,3,5,8]. YBCO shows a superposition of modulations localized near the $CuO_2$ bilayers, with basal plane components of their propagation vectors along both $a$ and $b$: $\mathbf{q}_a = (\delta_a,0,0)$ and $\mathbf{q}_b = (0,\delta_b,0)$ with correlation lengths up to $\xi_a \approx 70$ Å $\approx 20a$. Both $\mathbf{q}_a$ and $\mathbf{q}_b$ CDWs have ionic displacements perpendicular to the $CuO_2$ bilayers combined with

displacements parallel to these planes, which are $\pi/2$ out of phase[26]. These give rise to scattering along lines in reciprocal space given by $\mathbf{Q}_{CDW} = n\mathbf{a}^\star + m\mathbf{b}^\star + \ell\mathbf{c}^\star \pm \mathbf{q}_{a,b}$, where $n$ and $m$ are integers. The distribution of the scattered intensity along $\ell$ depends on the relative phase of the CDW modulations in the bilayers stacked along the $c$-direction. In zero magnetic field, there is weak correlation of phases in neighbouring bilayers and we observe scattered intensity spread out along the $\mathbf{c}^\star$ direction, peaked at $\ell \approx 0.5 - 0.6$. This is illustrated by our X-ray measurements on $YBCO_{6.67}$ ($P = 0.123$, $T_c = 67$ K and ortho-VIII CuO-chain ordering), shown in Fig. 1a,f. Note that the strong scattering around $\mathbf{Q} \sim (13/8,0,0)$ in Fig. 1a,b is due the CuO-chain ordering, which does not change with field, and can be subtracted, as in Fig. 1c,d. By taking cuts through the data, we obtain the intensity of the CDW scattering versus $\ell$ for the $\mathbf{q}_a$ and $\mathbf{q}_b$ positions (Fig. 1e,j).

### Field-induced anisotropic CDW correlations.
Figure 1 shows that the effect of applying a magnetic field is very different for two components ($\mathbf{q}_a$ and $\mathbf{q}_b$) of the CDW. For the $\mathbf{q}_a$ component of the correlations (Fig. 1b), the rod of scattering becomes stronger with no discernible change in the $\ell$ width or position of the maximum, that is, the correlations simply become stronger. In contrast, for the $\mathbf{q}_b$ correlations, (Fig. 1i) we see two qualitative changes. First, at $B \approx 10$ T, the rod of diffuse scattering becomes broader in $\ell$ and its peak position begins to move to larger $\ell$. Second, at $B \approx 15$ T, a new peak (shaded pink and first reported in ref. 24) appears centred on $\ell = 1$, but only for the $\mathbf{q}_b$ component. The new peak indicates that the sample has regions, where the CDW modulation is in phase in neighbouring bilayers and is coherent in three spatial directions. These regions would have a typical length along the $c$ axis of $\xi_c \approx 47$ Å.

### Structure of the three-dimensional CDW order.
We measured the intensity of the new three-dimensional (3D) CDW order in 14 different Brillouin zones. These data (Supplementary Note 3; Supplementary Table 1 and 2) are consistent with the high-field CDW structure of an individual bilayer being unchanged from that determined at zero field[26]. Both low- and high-field structures break the mirror symmetry of a bilayer, but in the high-field structure (Fig. 2a,d), the atomic displacements in adjacent bilayers are in phase. Thus, the high-field order has $\mathbf{q}_b = (0,\delta_b,0)$; however, its structure yields zero CDW intensity for $\ell = 0$ and nonzero for $\ell = 1$ positions (Supplementary Fig. 4). The relationship between the CDW structures at low and high field is to be expected, since the coupling between the two $CuO_2$ planes in a bilayer will be stronger than coupling with another bilayer. For the other basal plane direction, no CDW signal was found at $\mathbf{q} = (\delta_a,0,0)$ or $(\delta_a,0,1)$ for $B \leq 16.9$ T (Fig. 3c).

### The phase diagram and 3D CDW precursor correlations.
The $\ell$-dependent profiles in Fig. 1e,j contain information about the correlation between the phases of the CDW modulation in the bilayers stacked along the $c$ axis. For $B = 0$, the broad $\ell \approx 0.5 - 0.6$ peaks in Fig. 1e,j for $\mathbf{q}_a$ and $\mathbf{q}_b$ indicate that the CDW phase is weakly anti-correlated between neighbouring bilayers. On increasing the field above $B \approx 10$ T, the $\ell$-profile of the $\mathbf{q}_b$ correlations evolves. The onset of this evolution can be seen as an increase in the intensity of the scattering at $(0,4-\delta_b,1)$, see Fig. 4c, signalling the introduction of new $c$ axis correlations. This change is accompanied by a growth of correlations along the $b$ axis, as shown by the increase in the correlation length $\xi_{b,\ell=1}$ measured by the peak width of scans parallel to $\mathbf{b}^\star$ through the $(0,4-\delta_b,1)$

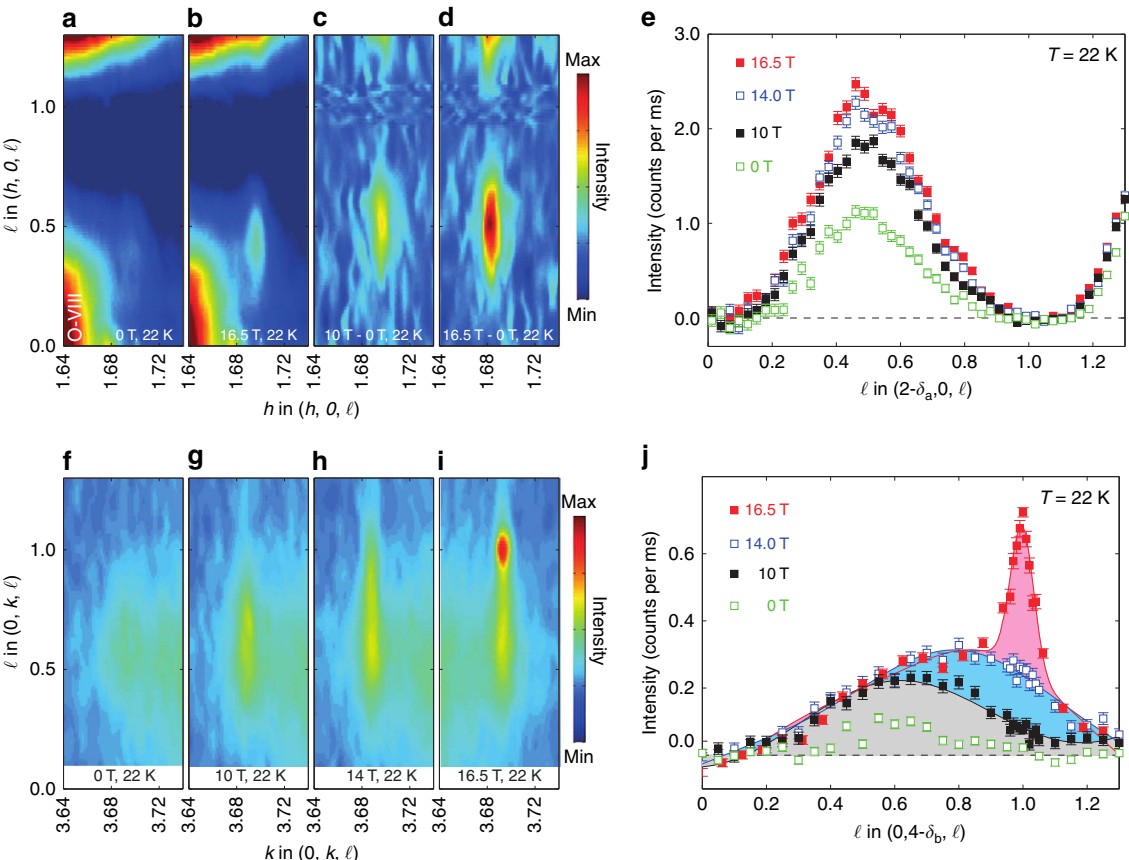

**Figure 1 | Charge density wave correlations induced by a magnetic field in YBa₂Cu₃O₆.₆₇.** A magnetic field applied along the $c$ axis introduces new CDW correlations propagating along both CuO bond directions $a$ and $b$ in the CuO₂ planes. (**a,b,f–i**) Raw X-ray scattering intensity data for the $(h,0,\ell)$ (**a,b**) and $(0,k,\ell)$ (**f–i**) planes for magnetic fields $0 \leq B \leq 16.5$ T. Strong features in (**a,b**) are due to CuO chain scattering. (**c,d**) Field-induced scattering for $(h,0,\ell)$. (**e,j**) CDW intensity along lines $\mathbf{Q} = n\mathbf{a}^\star + m\mathbf{b}^\star \pm \mathbf{q}_{a,b} + \ell\mathbf{c}^\star$ isolated from data such as (**a,b,f–i**). The CDW intensity has been isolated by fitting peaks due to the CDW and other structural features to a series of $h$- or $k$-cuts through data such as (**a,b,f–i**). CDWs propagating along the $a$ axis (**a–e**) within individual bilayers become stronger without changing phase relationship with neighbouring bilayers. Those propagating along $b$ axis (**f–j**) become in phase with neighbouring bilayers, which changes the profile in $\ell$. The shaded areas in (**j**) show: weakly anti-correlated CDW (grey); 3D CDW precursor correlations (blue); and 3D CDW order (red). Error bars are s.d.'s determined by counting statistics. We describe reciprocal space as $\mathbf{Q} = h\mathbf{a}^\star + k\mathbf{b}^\star + \ell\mathbf{c}^\star$, where Ideally mod $(\mathbf{a}^\star) = 2\pi/a$, $a = 3.81$ Å, $b = 3.87$ Å and $c = 11.72$ Å.

position (Fig. 4e). We describe this state as 3D CDW precursor correlations. The onset temperature $T \approx 65$ K of the precursor correlations at high field ($B = 16.5$ T) may be determined from the increase in $\xi_{b,\ell=1}$ and the scattering intensity at the $(0,4-\delta_b,1)$ position (Fig. 4c,e). This allows us to designate a region of the $B - T$ phase diagram (Fig. 5).

At higher fields, $B \gtrsim 15$ T, a peak (shaded pink in Fig. 1j) develops abruptly in the $\ell$-profile at $\ell = 1$. The abrupt onset of the peak signals a rapid growth of the $c$ axis correlation length $\xi_c$ (Fig. 4d,e). The growth of correlations in one spatial direction followed by growth in a second direction is typical of systems, with anisotropic coupling. Another CDW system that shows this behaviour[27] is NbSe₃. Large correlated regions develop first in planes, where the order parameter is most strongly coupled. These act to amplify the coupling in the remaining direction. In case of YBCO₆.₆₇, the in-plane correlation length continues to grow down to low temperatures with $\xi_{b,\ell=1} = 80b = 310$ Å (at ~10 K and 16.5 T). The $c$ axis correlation length, however, saturates with $\xi_{c,\ell=1} = 47$ Å at $T \approx 30$ K. All these changes together signal the transition to a new phase (see Fig. 5 pink region), which we label 3D CDW order identified with a phase transition also seen in ultrasound[16] and thermal Hall effect[17] measurements. At the lowest temperatures, $T \lesssim 25$ K, we observe (Fig. 4a) a suppression of the 3D CDW peak intensity signalling a

competition between the superconducting and 3D CDW order parameters.

Previous X-ray[5,8] and NMR[25] measurements on YBCO₆.₆₇ have shown that the weak anti-phase ($\ell = 1/2$) CDW correlations appear at $T \approx 150$ K. Further NMR anomalies in the form of line splittings[11,15] are observed at $T \approx 65$ K for $B = 28.5$ T and at $B \approx 10$ T for $T = 2$ K. These anomalies that are displayed on Fig. 5 appear to coincide with the onset of the 3D precursor correlations reported here. The fact that NMR sees similar transitions shows that the 3D CDW precursor correlations we observe are static on timescales $\tau \gtrsim 0.1$ ms. Correlations that are static[25] and short ranged are necessarily controlled by pinning with quenched disorder playing a role.

**Doping dependence.** We also studied other dopings of YBCO₆₊ₓ. For YBCO₆.₆₀ with hole doping $P = 0.11$ and ortho-II oxygen chain structure, a very similar onset field (Fig. 5) and $c$ axis correlation length $\xi_c$ were found. In YBCO₆.₅₁ and YBCO₆.₇₅, no 3D order was observed for $B \leq 16.9$ T (Fig. 3a). However, we do observe the precursor movement of the CDW scattering to higher $\ell$ implying that this structure is likely to appear at higher fields. Thus, the 3D order is most easily stabilized for doping around $p = 0.11$–$0.12$ (Fig. 5b).

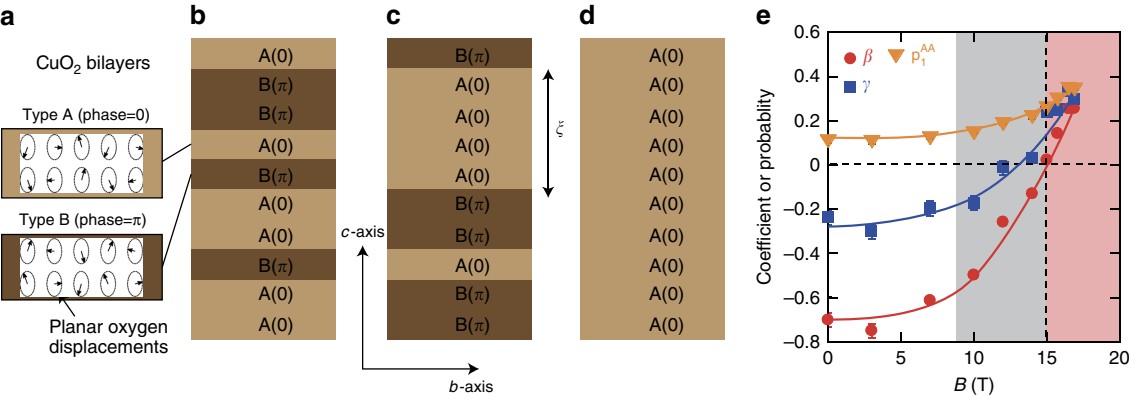

**Figure 2 | Magnetic field effect on *c* axis correlations in YBa$_2$Cu$_3$O$_{6.67}$.** (**a**) Schematic of the CDW modulation in a bilayer[26]. Arrows represent the displacement of the planar oxygens. Two phases: A ($\theta = 0$), B ($\theta = \pi$) of the modulation are shown. (**b–d**) Representative CDW stacking sequences for different fields. (**b**) ($B = 0$) Weakly anti-correlated and (**c**) ($B = 16.5$ T) short-range three-dimensional (3D) order with correlated regions of size $\xi_{(c)} \approx 4c$. (**d**) Weakly pinned fully 3D coherence (large B). (**e**) Field-dependent parameters determined from fitting data such as Fig. 1j to the Markov model described in Methods: nearest-neighbour coupling ($\beta$); next-nearest-neighbour coupling ($\gamma$); and the proportion of bilayer AA pairs separated by one lattice spacing ($P_1^{AA}$). Note $P_1^{BB} = P_1^{AA}$. Errors are determined from least square fitting of model to data.

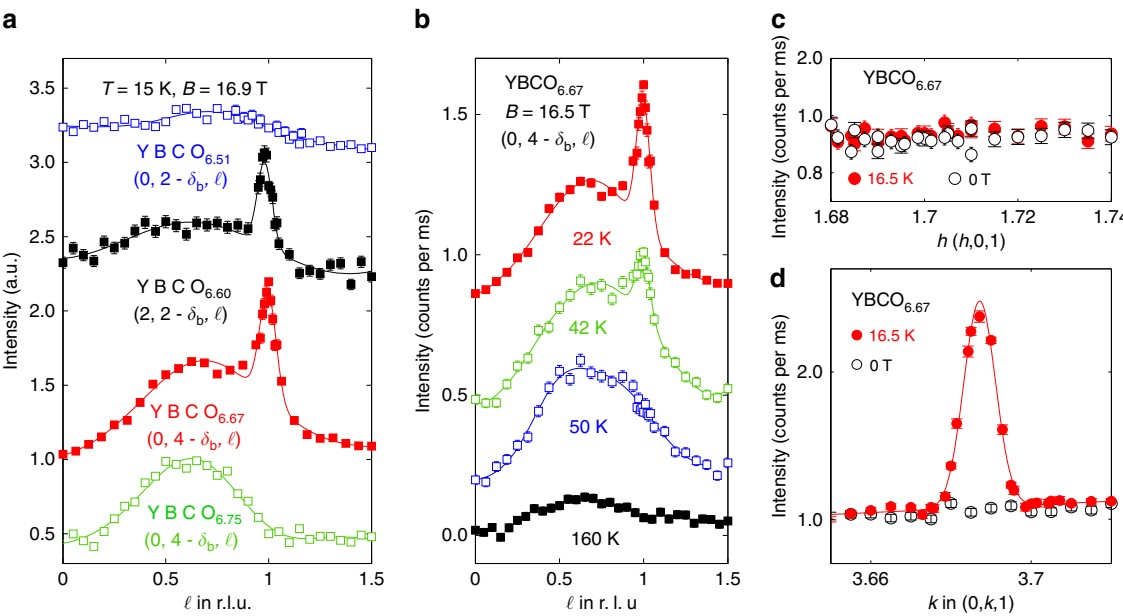

**Figure 3 | Doping temperature and field dependence of induced CDW correlations.** (**a**) $\ell$-Scans along $(0, \delta_b, \ell)$ for different dopings of YBCO at $T \approx 8$ K and $B = 16.9$ T showing the field-induced 3D correlations or lack thereof. (**b**) Temperature dependence of $\ell$-scans along $(0, \delta_b, \ell)$ measured on YBCO$_{6.67}$ at $B = 16.5$ T. All curves in (**a,b**) have been given an arbitrary shift. (**c,d**) h-Scans through $(2-\delta_a, 0, 1)$ and k-scans through $(0, 4-\delta_b, 1)$ in YBCO$_{6.67}$ for zero field and $B = 16.5$ T. Error bars are s.d.'s determined by counting statistics.

## Discussion

A feature of the present data is that the $\ell$-dependent profiles measured along $(0, \delta_b, \ell)$ and their field evolution (for example, Figs 1j and 3a,b) cannot be understood as a superposition of broadened peaks centred at $\ell = 1/2$ and $\ell = 1$. The change in the $\ell$-dependence of the intensity represents a variation with field of the stacking of the bilayer CDWs. To interpret these profiles, we use a simple statistical approach based on a Markov chain (Methods; Supplementary Note 2) to model possible CDW stacking sequences along the *c* axis and compute the scattering profile as a function of $\ell$. A good description of our data is obtained if we assume that the CDW phase difference between neighbouring bilayers is 0 or $\pi$ (Supplementary Note 1 and 2; Supplementary Fig. 2). The parameters in our model are the nearest- and next-nearest-neighbour couplings $\beta$ and $\gamma$, where positive values favour the coupled bilayers having the same phase. At $B = 0$, the model shows that the broad $\ell \approx 0.5 - 0.6$ peaks in Fig. 1e,j are due to weakly anti-correlated bilayers (Fig. 2b,e). The field evolution of the $\ell$-dependent profiles for $\mathbf{q}_b$ (Fig. 1j), including the formation of the $\ell = 1$ peak, may be modelled by a continuous variation of $\beta$ and $\gamma$ from anti-phase coupling at low field to same-phase coupling at high field (Fig. 2e). The sign of $\beta$ changes near the onset of the 3D order at $B \approx 15$ T. Thus, we find that a *c* axis magnetic field can control the coupling between the CDWs in neighbouring bilayers. The field control of the coupling most likely arises through the suppression of superconductivity by field. Magnetic field strengthens the correlations along the *a* axis (Fig. 1e); however, it does not increase correlation lengths.

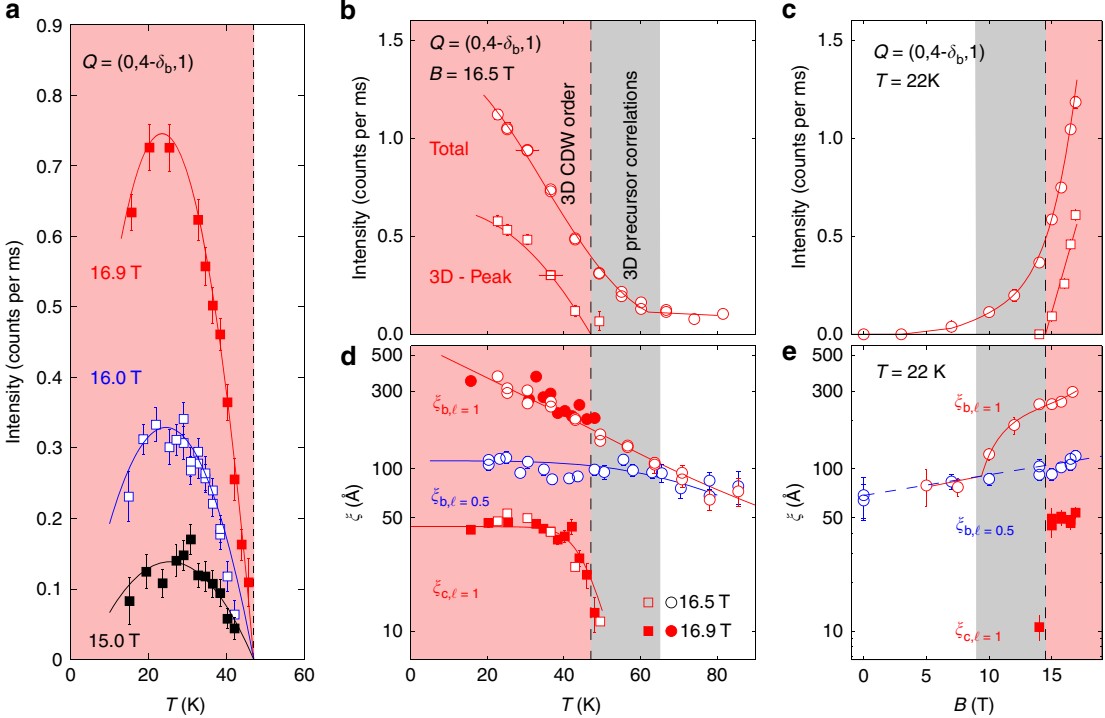

**Figure 4 | Evolution of charge density wave correlation lengths and intensities with magnetic field and temperature in YBa$_2$Cu$_3$O$_{6.67}$.** (**a**) Intensity of the 3D CDW peak extracted from $\ell$-scans through **Q** = (0,4-$\delta_b$,$\ell$) versus temperature at fields as indicated. (**b,c**) Total CDW intensity determined from $k$-scans (open circles) and 3D CDW peak intensity determined from $\ell$-scans (closed squares) through the (0,4-$\delta_b$,1) position. (**d,e**) Correlation lengths $\xi_{b,\ell=1}$, $\xi_{c,\ell=1}$, $\xi_{b,\ell=1/2}$ determined from the resolution-corrected peak widths ($\sigma = \xi^{-1}$) of scans through (0,4-$\delta_b$,1), (0,4-$\delta_b$,1) and (0,4-$\delta_b$,1/2) positions, respectively. The saturation of $\xi_{c,\ell=1} = 47$ Å and $\xi_{b,\ell=1/2} \sim 100$ Å is likely related to disorder even though it has been shown that $\xi_{b,\ell=1/2}$ is insensitive to oxygen disorder[34]. Error bars are s.d.'s of the fit parameters described in the Methods.

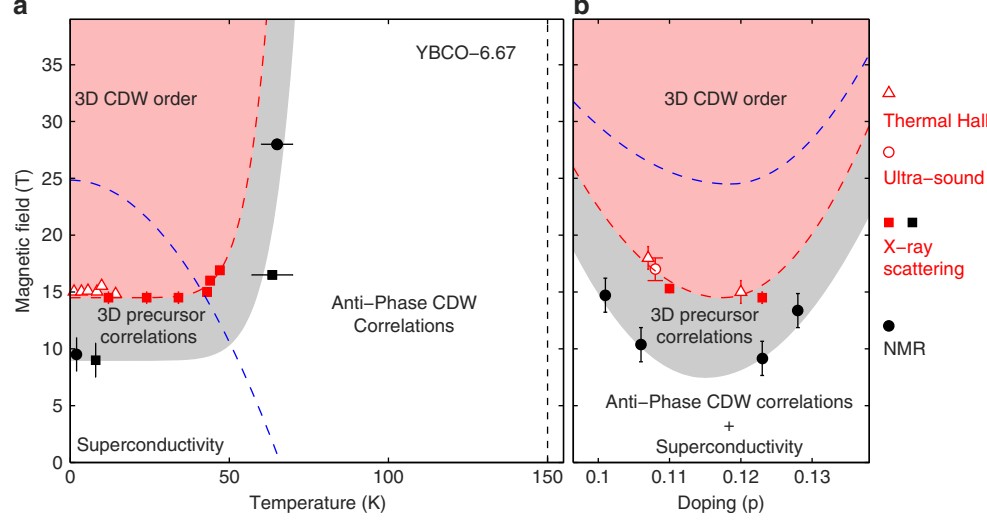

**Figure 5 | Phase diagram of YBa$_2$Cu$_3$O$_{6+x}$.** The pink shaded areas represent the regions where short-range 3D CDW order exists. Grey bands indicate the regions where growing 3D CDW precursor correlations are observed. (**a**) Temperature-magnetic field phase diagram. (**b**) Doping-magnetic field phase diagram. Solid red square points indicate the onset of a 3D CDW order with **q**$_b$ = (0,$\delta_b$,0) determined from the variation of the $\xi_{c,\ell=1}$ correlation length and the intensity of the 3D peak (Fig. 4). Triangles are the Fermi surface reconstruction onset determined from thermal Hall coefficient[17]. Solid black squares indicate the onset of growing in-plane CDW correlation lengths (3D precursor correlations) determined from the variation of $\xi_{b,\ell=1}$ (Fig. 4d,e). Dashed blue lines in (**a,b**) indicate $B_{c2}$ line[35]. Solid black circles in (**a,b**) are derived from NMR[11,15]. The vertical black dashed line is the onset of weakly anti-phase CDW correlations (refs 4, 5 and 8). Red circular and triangular points originate from ultrasound[16] and thermal Hall effect[17] experiments, whereas the red squares are the field onset of **q**$_b$ = (0,$\delta_b$,0) found by X-ray diffraction.

Possible explanations for this difference in behaviour include the influence of the CuO chains promoting the $b$ axis modulations or the chains pinning the $a$ axis CDW modulations.

We conclude that the appearance of 3D CDW order corresponds to the onset of new $c$ axis electronic coherence and hence electronic reconstruction. This is supported by thermal

**Table 1 | YBCO samples investigated in this study.**

| x in YBCO | Oxygen ordering | Doping level p | $T_c$ (K) | $B_c$ (T) | $\xi_b$ (b) | $\xi_c$ (c) |
|---|---|---|---|---|---|---|
| 6.51 | o-II | 0.096 | 59 | >16.9 | — | — |
| 6.60 | o-II | 0.11 | 61.8 | 15.3 ± 0.35 | 48 | 4.5 |
| 6.67 | o-VIII | 0.123 | 67 | 14.5 ± 0.5 | 80 | 4 |
| 6.75 | o-III | 0.132 | 74 | >16.9 | — | — |

YBCO, $YBa_2Cu_3O_{6+x}$.
The correlation length $\xi_b$ and $\xi_c$ of the $(0, \delta_b, 0)$ CDW order at the highest measured fields and lowest temperature are given in units of the lattice parameters $b = 3.87$\AA(Å) and $c = 11.7$ Å.

Hall conductivity measurements[17] that demonstrate Fermi surface reconstruction at the same field (Fig. 5a). At highest fields investigated, $B = 16.9$ T, the structure of the CDW within individual bilayers involves the same breaking of mirror symmetry observed at zero field[26], which has been posited to lead to Fermi surface reconstruction[28,29].

## Methods

**Experimental details.** Our experiments used 98.5 keV hard X-ray synchrotron radiation from the PETRA III storage ring at DESY, Hamburg, Germany. A 17 T horizontal cryomagnet[30] was installed at the P07 beamline. Access to the $(h, 0, \ell)$ and $(0, k, \ell)$ scattering planes was obtained by aligning either the $a$–$c$ axes or the $b$–$c$ axes horizontally, with the $c$ axis approximately along the magnetic field and beam direction. The samples were glued to a pure aluminium plate on which was mounted a Cernox thermometer for measurement and control of temperature. With the high intensities of PETRA III, a small amount of beam heating of the sample was observed. By observing the effect of changes in beam heating (controlled by known attenuation) on the measured temperature of the 3D phase transition, we determined the effect of the beam on the sample temperature near 40 K. The sample heating at other temperatures was determined using the Cernox thermometer and a model of the heat flow from the sample to the aluminium plate. We estimate that there is an absolute uncertainty in our temperature determination of ± 2 K. The relative temperature uncertainty is smaller than this.

Four YBCO crystals with different in-planar doping and different oxygen chain structure were studied (Table 1). Except for the YBCO$_{6.60}$ sample, detailed descriptions of these crystals are found in refs 5,31,32. The YBCO$_{6.60}$ sample was studied with the scattering plane defined by $(k, k, 0)$ and $(0, 0, \ell)$. This configuration has the advantage that CDW modulations along both $a$ and $b$ axis directions could be accessed without reorienting the sample. The absence of $\ell = 0,1$ CDW order along the $a$ axis direction was checked using the $(h, 0, \ell)$ scattering plane.

**Data analysis.** $h$- and $k$-scans, as shown in Fig. 3c,d, are fitted with a single Gaussian function on a weakly sloping background. $\ell$-scans with a well-defined peak at $\ell = 1$ (Figs 1j and 3a,b) are fitted using a two Gaussian functions. Correlation lengths $\xi = 1/\sigma$ are defined by the inverse Gaussian s.d. $\sigma = (\sigma_{meas}^2 - \sigma_R^2)^{0.5}$. The instrumental resolution $\sigma_R$—for a CDW reflection—was estimated at Bragg reflections near to the measured CDW reflections. Resolution-corrected correlation lengths are given in Table 1.

**Simulation of scattering profiles.** We use a simple Markov chain model[33] of order $m = 2$ to interpret the diffuse and $\ell = 1$ scattering profiles, for example, in Fig. 1j. A Markov chain is a stochastic series. Here we generate a series of two types of bilayer (A and B) corresponding to the phase of the displacement of the CDW in a given bilayer. We represent the bilayer type at position index $i$ along the $c$-direction by a stochastic variable $x_i$. This can take either the value $x_i = 1$, denoting bilayer type A, or $x_i = 0$ for type B. We create a series of bilayers starting, for instance, with an A bilayer, followed by a B. The probability $P(x_i = 1)$ of adding an A bilayer at position $i$, preceded by $x_{i-1}$ and $x_{i-2}$, in the series is given by:

$$P(x_i = 1 \mid x_{i-1}, x_{i-2}) = \alpha + \beta x_{i-1} + \gamma x_{i-2}. \quad (1)$$

Clearly, $P(x_i = 0) = 1 - P(x_i = 1)$. Equation (1) is a recipe, using random numbers to represent the probabilities, to create a series of bilayers with a given amount of correlation built in. Let $m_A$ ($m_B = 1 - m_A$) be the fraction of A-type (B-type) bilayers in the series and $P_1^{AA}$ the proportion of AA bilayer pairs separated by one lattice spacing. We choose $\alpha = \frac{1}{2}(1 - \beta - \gamma)$ so that macroscopically $m_A = m_B = \frac{1}{2}$. A number ($N \sim 500$) of stochastic series $x_i$ ($N_{site} = 100$) subject to a given $\alpha$, $\beta$ and $\gamma$ are generated. The corresponding scattered intensity (assuming the single-unit cell structure from ref. 26) for each series is calculated and averaged. $\beta$ and $\gamma$ are adjusted to give the best fit to the data and $P_1^{AA}$ is calculated.

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

## Acknowledgements

We wish to thank M.H. Julien and L. Taillefer for helpful discussions. This work was supported by the Engineering and Physical Sciences Research Council (EPSRC) (grant numbers EP/G027161/1, EP/K016709/1 and EP/J015423/1), Danish Agency for Science, Technology and Innovation through DANSCATT and grant number 0602-01982B and the Swiss National Science Foundation grant number BSSGI0-155873.

## Author contributions

R.L., D.A.B and W.H. grew and prepared the samples. J.C., E.B., M.H., N.B.C., U.R., M.Z., E.M.F. and S.M.H. conceived and planned the experiments. J.C., E.B., O.I., A.H., M.Z., E.M.F. and S.M.H. carried out the experiments. J.C. and S.M.H. carried out the data analysis and modelling. All co-authors contributed to the manuscript.

## Additional information

**Competing financial interests:** The authors declare no competing financial interests.

