## [Peer Review file · Nature Communications]

Transferred manuscripts:

Reviewers' Comments:

Reviewer #1 (Remarks to the Author)

I have read the revised version of this manuscript and I am satisfied with the authors responses and changes. I recommend that this paper be published as it is.

Reviewer #2 (Remarks to the Author)

As noted in my original report, this paper provides a detailed and valuable study of the magnetic field dependence of CDW order in YBCO, elucidating previous work and providing an important reconciliation with between x-ray scattering and other probes such as NMR and ultrasound. This is sure to have impact for those working on CDW order in the cuprates and will make an important contribution to the field. The manuscript is well written, the measurements are of high quality and the authors have appropriately addressed all concerns from my previous report. As such, I have no hesitation in recommending the paper for publication in Nature Communications.

Reviewer #3 (Remarks to the Author)

As I said in my previous report, this is a tour de force study. I had also recommended at that time the paper be published in Nature Communications. I am glad the authors have taken my suggestion and submitted to Nature Communications. Despite the other reports on this paper, I am fine with the present version being published.